# Blindness and the Reliability of Downwards Sensors to Avoid Obstacles: A Study with the EyeCane

**DOI:** 10.3390/s21082700

**Published:** 2021-04-12

**Authors:** Maxime Bleau, Samuel Paré, Ismaël Djerourou, Daniel R. Chebat, Ron Kupers, Maurice Ptito

**Affiliations:** 1École D’optométrie, Université de Montréal, Montréal, QC H3T 1P1, Canada; maxime.bleau.1@umontreal.ca (M.B.); samuel.pare.1@umontreal.ca (S.P.); ismael.djerourou@umontreal.ca (I.D.); endogonidia@gmail.com (R.K.); 2Visual and Cognitive Neuroscience Laboratory (VCN Lab.), Department of Psychology, Faculty of Social Sciences and Humanities, Ariel University, Ari’el 40700, Israel; danielc@ariel.ac.il; 3Navigation and Accessibility Research Center of Ariel University (NARCA), Ari’el 40700, Israel; 4Department of Neuroscience, University of Copenhagen, 2200 Copenhagen, Denmark

**Keywords:** navigation, blindness, sensory substitution, EyeCane, obstacle detection, avoidance, collision

## Abstract

Vision loss has dramatic repercussions on the quality of life of affected people, particularly with respect to their orientation and mobility. Many devices are available to help blind people to navigate in their environment. The EyeCane is a recently developed electronic travel aid (ETA) that is inexpensive and easy to use, allowing for the detection of obstacles lying ahead within a 2 m range. The goal of this study was to investigate the potential of the EyeCane as a primary aid for spatial navigation. Three groups of participants were recruited: early blind, late blind, and sighted. They were first trained with the EyeCane and then tested in a life-size obstacle course with four obstacles types: cube, door, post, and step. Subjects were requested to cross the corridor while detecting, identifying, and avoiding the obstacles. Each participant had to perform 12 runs with 12 different obstacles configurations. All participants were able to learn quickly to use the EyeCane and successfully complete all trials. Amongst the various obstacles, the step appeared to prove the hardest to detect and resulted in more collisions. Although the EyeCane was effective for detecting obstacles lying ahead, its downward sensor did not reliably detect those on the ground, rendering downward obstacles more hazardous for navigation.

## 1. Introduction

While navigating in an environment, humans rely on visual information to identify obstacles, evaluate distances, and create a mental map of their surroundings. Thus, the loss of visual input as occurs in blindness has a proportionally greater effect on navigational abilities and independence as compared to other senses [1]. To overcome this deficit in safe locomotion, blind individuals must learn to use many aids and tools to obtain environmental information that is necessary for diverse everyday tasks such as wayfinding and circumventing obstacles [2]. As the most widespread of these mobility aids, the white cane functions as an extension of the hand and arm to enable obstacle detection and to furnish information about ground textures and level changes (i.e., drop-offs, steps, and curbs) encountered during locomotion. However, the white cane does not generally provide information about obstacles positioned higher than the user’s pelvis, thus leaving the upper body at risk of collisions and injuries [3,4] (Figure 1A). This significant limitation considerably impedes safety, which can be discouraging to blind individuals and ultimately leads to social isolation [5]. There is therefore a pressing need to develop and explore new technologies that could potentially improve their safety and autonomy in daily travels. One promising area of research is sensory substitution, which aims to convey visual information to blind individuals through touch and sound with sensory substitution devices (SSD) [6]. Many different SSDs have been designed to substitute for vision, while others provide guidance that is more task-based, meaning that they offer a specific aspect of visual information (e.g., color, shape, distance or letters) strictly pertinent to preforming a certain task (e.g., wayfinding, obstacle circumvention, or reading).

For instance, the tongue display unit (TDU) is a tactile-to-vision SSD that, in a laboratory environment, allows blind individuals to discriminate shapes [7,8], movement [9], pathways [10], and letters [11,12], while enabling users to detect and avoid obstacles in a life-size obstacle course [13]. However, the visually impaired community has not adopted the TDU, since the abundant information given by the device is complex and requires the user to attend constantly to maintain an adequate level of performance, which quickly leads to exhaustion (see cognitive load problem in [6,14,15]). Therefore, devices like minimalist SSDs and electronic travel aids (ETA) that convey a simpler signal could maintain safety while limiting fatigue [2,6].

ETA devices are equipped with sensors (ultrasonic, infrared, or electromagnetic), radars, or cameras to capture information about the environment and convey encoded signals to blind users by tactile (vibration) or auditory stimulation [2,16,17,18]. While ETAs fall within the general category of SSDs, they mostly provide simple feedback that is strictly relevant for navigation, such as the presence of obstacles in the path of travel, and their distance and location. Since the 20th century, researchers have developed multiple ETAs and other SSDs as mobility aids to assist blind individuals [2]. These devices generally qualify as either primary tools (that can be used independently from the white cane) [19] or secondary tools (that must be used in conjunction with the white cane) [20,21]. However, the broader blind population has not adopted such aids due to many factors such as training requirements, high cost, and low portability [6,22].

To circumvent these shortcomings, the EyeCane was developed as a torch-like ETA equipped with narrow field of view (FOV) infrared sensors that capture distance information about obstacles and convey it to the user’s hand through vibrations. This gives the possibility to “feel” the immediate environment without touching it directly. Several studies that evaluated the EyeCane’s potential in rehabilitation have concluded that its users were able easily to estimate distances, navigate, and detect and avoid obstacles in real [23] and virtual [24,25] environments after only a brief training.

In a recent study [26], the EyeCane was adapted with two narrow FOV infrared sensors with a 1.5 m sensing range (Figure 1B) to test its reliability as a primary and secondary aid to tackle the safety challenge posed by waist-up obstacles. The authors of that study stated the hypothesis that the EyeCane might suffice as a reliable primary mobility aid if it were equipped with a third sensor pointing toward the ground (Figure 1D), thus replacing the need for a physical cane for detecting foot-obstacles. The goal of the present study was to investigate the reliability of such a downward sensor. For this purpose, we used a single narrow FOV infrared sensor with a 2 m sensing range pointed toward the ground (Figure 1C), and we assessed its reliability in a life-size obstacle course presenting risks of collision, tripping, and falling. Furthermore, we compared the EyeCane navigational capacities of late (LB)- and early-blind individuals (EB) to blindfolded sighted controls (SC). We hypothesized that smaller obstacles at ground level would prove more difficult to detect for the three groups of subjects, and that EB would be more adept than the two other groups at learning to navigate the obstacle course.

**Figure 1 sensors-21-02700-f001:**
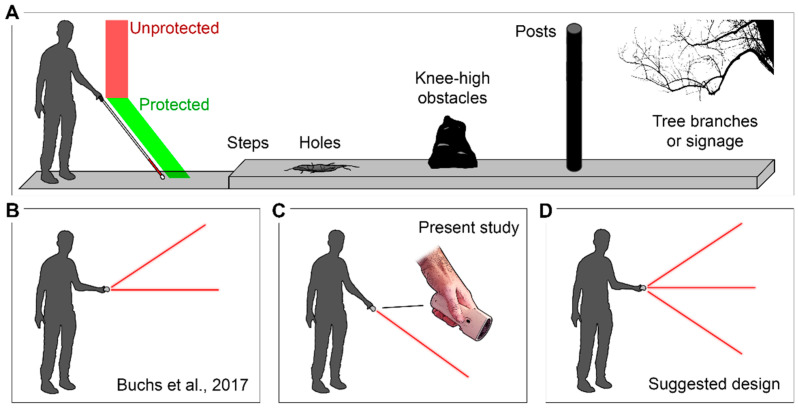
Schematics of (**A**) an individual with the white cane facing daily-life obstacles, the cane protects him from low obstacles; however, the individual is at risk of head injury with hanging obstacles like tree branches and signage; (**B**) an individual handling the EyeCane design used in [26]; (**C**) our experimental design with the single-sensor EyeCane in a downward directed manner and (**D**) an individual handling the three-sensors EyeCane design suggested in [26].

## 2. Materials and Methods

### 2.1. Participants and Ethics

Participants were recruited in Montreal (Canada) through the database of the Harland Sanders Research Chair in Vision Neuroscience, and in Denmark through the database of the BRAINlab at the University of Copenhagen. The experiment took place at the School of Optometry at the Université de Montréal and the University of Copenhagen. Ten EB (mean age: 44 ± 12 years; 2 females and 8 males) and nine LB (mean age: 48 ± 14 years; 4 females and 5 males) were recruited in this study (Table 1).

We also included 10 sighted controls (SC) with normal vision (mean age: 44 ± 13 years; 4 females and 4 males), who were age- and sex-matched to the blind participants. SC were blindfolded throughout the experiment. All blind participants were expert users of the white cane, and two had a guide dog. To evaluate the influence of experience-dependent plasticity in the LB group, we calculated the blindness duration index (BDI) according to the formula “(age–age onset blindness)/age” (as described in [27]). The BDI score can range from 0 to 1, expressing the proportion of his life that a person has been blind, with low scores indicating recent onset of blindness and high scores longer duration of blindness. The average BDI was 0.42 ± 0.20 (range: 0.03 to 0.64) while the mean onset of blindness was 28.2 ± 13.7 years. Participants had no associated neuropathy that could affect their navigational performance or mental representation. Before starting the experiment, participants completed a questionnaire regarding their blindness and spatialization abilities and signed a consent form. The protocol was approved by the Clinical Research Ethics Committee of the University of Montreal (CERC-19-097-P) and by the local ethics committee of the University of Copenhagen (Region Hovestaden; Protocol nr: H-6-2013-004) and was conducted in accordance with the Declaration of Helsinki.

### 2.2. Apparatus

The EyeCane is a small (4 × 5 × 13 cm) and light-weight (∼100 g) hand-held mobility device with a form similar to a flashlight (Figure 1 and Figure 2) and a long battery life (up to 24 h use, simple to charge) [23]. Equipped with an infrared emitter and sensor (Sharp GP2Y0A02YK0F), it emits a narrow light beam (<5°) in the direction at which it is aimed and detects the reflected signal. The EyeCane then determines the distance to the hit object and translates the information into vibration, encoded with varying intensity levels. Therefore, when an obstruction is detected at a range of 20 to 150 cm, the device vibrates in the user’s palm, and its intensity is inversely proportional to the distance of the obstacle: the closer the obstacle, the higher the intensity of the vibration. The full specification details have been described previously [23], and the sensors were previously shown to work on different materials and in various lighting conditions [23,28,29].

### 2.3. Experimental Procedure

The experiment consisted of three parts: the training phase, the test phase, and the post-test phase, when the participants’ experience and level of satisfaction were assessed.

#### 2.3.1. Obstacle Course

The experiments were conducted in a life-size obstacle course simulating daily-life situations encountered during outdoor and indoor travel. This obstacle course consisted of a hallway measuring 21 m long and 2.4 m wide, equipped with differently sized obstacles to evaluate detection, avoidance, and identification performances with the use of the EyeCane as a standalone aid (Figure 2). To minimize the risk of injury, all obstacles were constructed from cardboard and foam.

Four types of obstacles were designed for the experiment. These included “cube” (height: 0.61 m; width: 0.45 m; depth: 0.45 m), “door frame” (height: 1.88 m; depth: 0.45 m; door width: 0.71 m; total width: 2.4 m), “step” (height: 0.15 m; depth: 0.15 m; width: 2.4 m), and “post” (height: 1.45 m; diameter: 0.10 m) (Figure 2). These four types of obstacles were chosen to represent a range of scenarios with differences in floor surfaces and other obstacles in the path of travel. The “cube” represented large knee-high obstacles; the “door frame” represented narrow passages such as door frames and any space between two obstacles or between two other pedestrians; the “step” represented possible changes in the floor denivelation (i.e., steps, curbs, and sidewalks); and the “post” represented thin obstacles often found on the sidewalk such as signage.

#### 2.3.2. Training Phase

All participants underwent the same training procedures. They were first verbally introduced about the EyeCane’s principle of operation and then familiarized with the device in an otherwise empty room containing a single obstacle, i.e., a cardboard tower (0.4 × 0.4 × 2 m). This familiarization phase served to introduce the concepts of obstacle recognition, size, and distance estimation, while being guided and unguided by the experimenters. Participants were taught that object recognition is possible by scanning the object with the device (side-to-side and up–down) and detecting its edges to gain information about its shape. The participants then underwent a simulated detection and avoidance task in the experimental walkway (21 × 2.4 m), with placement of three cardboard towers at 3 m apart on the longitudinal axis and randomly positioned on the horizontal axis.

The participants were then introduced to the scanning technique used for the experiment. They were taught how to scan the environment with the device pointed toward the ground in front of them such that they felt a constant, low-intensity vibration emitted from the device. This technique ensures that users always detect the ground and are thus alerted to any changes in the floor surface or the space in front of them. In fact, the arm movements required for this technique were closely related to the “two-point touch technique” that blind individuals learn in the context of white cane orientation and mobility (O&M) lessons [30,31]. The goal of this specific technique was to simulate the use of a third sensor (pointed toward the ground) while isolating it from the two other sensors’ signals. Therefore, our experimental design allows us to assess the reliability of this specific sensor without any added complexity and, thus, to assess the suggested three-sensors EyeCane’s suitability as a primary mobility aid.

Finally, the participants were familiarized with the four types of obstacles used in the experiment. During this phase, the participants used the scanning technique and were guided by the experimenter toward each obstacle. They were taught to detect and identify each of the four types of obstacles. The complete training phase (device familiarization, scanning technique, and obstacles familiarization) averaged 15 min for most participants.

#### 2.3.3. Test Phase

All participants navigated the same 12 configurations of the obstacle test in random order. For each configuration, six obstacles of random type were placed as in Figure 2, thus comprising a total of 18 encounters for each obstacle.

The assigned task was to cross the corridor as quickly as possible while detecting, identifying, and avoiding obstacles. The participants were monitored by two experimenters for safety reasons and data collection. Object *detection, identification* and *avoidance* are distinct processes that follow each other sequentially and serve the same goal: ensuring the individual’s safety during navigation. Detection is the first step toward attaining the navigation goal as it allows the individual to gain awareness about the presence of an obstacle in the path of travel, to adjust his/her pace for safety, and to anticipate contact or avoidance, thus decreasing the risk of a dangerous collision. The *identification* process occurs when the individual gains information about the nature and dimensions of an object. *Avoidance* is the culmination of both processes, occurring when the individual has to plan a deviation in his/her path of travel according to the obstacle’s position and dimensions, to successfully execute this new path, and finally to regain the initial track. Therefore, to serve effectively as a primary mobility aid, the EyeCane must prove reliable for obstacle detection, identification, and avoidance, serving for each type of obstacles, especially those at ground level (i.e., the task’s “step” obstacle, potholes, and curbs). Another important factor for efficiency and reliability is the amount of time needed to detect, identify, avoid obstacles and complete the course; thus, we also measured crossing time.

### 2.4. Statistical Analysis

The collected data consisted of the average crossing time and indices of three types of performance: obstacle detection, avoidance, and identification. Since avoidance performance also counted those obstacles that participants did not encounter (being too small or too peripheral to their track), we therefore separately evaluated participants’ performances in avoiding detected obstacles. Collision data was also calculated as the opposite scalar of avoidance performance.

Data were analyzed using JASP, an open-source graphical program for statistical analysis developed by the University of Amsterdam [32]. Two-way ANCOVA tests corrected for age and sex, or the nonparametric equivalent Kruskal–Wallis test, were used to determine the effect of group (EB, LB, and SC) on time and performance data. We then verified the effects of obstacle type (“cube”, “door frame”, “step”, and “post”) and group (EB, LB, and SC) on detection, as well as their interaction on collision rates, using two-way ANCOVA corrected for age and sex. *Post hoc* T-tests (or Mann–Whitney) with Bonferroni correction were performed to identify any significant differences. Results are presented as mean ± SD.

## 3. Results

*Crossing time*: In terms of the average time to cross the obstacle course through the 12 trials, EB were faster with 154.5 ± 39.6 s than LB with 240.7 ± 117.2 s and SC with 304.9 ± 143.8 s (Figure 3A). The ANCOVA corrected for age and sex test indicated that there was a statistically significant difference in crossing times between groups (F(2,22) = 7.290, *p* = 0.004, η^2^ = 0.384) but failed to show an effect of age or sex (*p* > 0.05). Post hoc comparison with Bonferroni correction revealed that EB were significantly faster than SC (*p* = 0.003), but no differences were found between EB and LB or between LB and SC (*p* > 0.05).

*Obstacle detection, identification, and avoidance*: the EB group detected 54.9 ± 8.3% of the obstacles, whereas LB detected 45.2 ± 11.3% and SC 51.1 ± 14.1% of the obstacles. A two-way ANCOVA with correction for age and sex did not indicate any significant differences between groups in detection rate (F(2,24) = 1.972, *p* = 0.428, η^2^ = 0.070). For obstacles that were successfully detected, the analyses did not find significant group differences for identification rate (F(2,22) = 0.452, *p* = 0.642, η^2^ = 0.039). EB identified 88.4 ± 11.3% of the detected obstacles, LB 81.4 ± 10.6% and SC 81.6 ± 16.1%. Moreover, EB avoided 86.3 ± 11.8% of detected obstacles, LB 87.7 ± 7.2% and SC 68.3 ± 15.4%. Since the Shapiro–Wilk test signaled a departure from normality in avoidance scores, we applied the nonparametric Kruskal–Wallis test for this contrast. The analysis indicated a statistically significant difference between the groups (*H*(2) = 6.844, *p* = 0.033, η^2^ = 0.336). Mann–Whitney post hoc tests with Bonferroni correction showed a significant advantage in obstacle avoidance for the EB (*p* < 0.01) and LB groups (*p* < 0.01) compared to SC, but no difference between EB and LB (*p* > 0.05) (Figure 3B). No effect of age nor sex was found in any of the analyses. For the LB group, there was no significant correlation with BDI in any of the measures (*p* > 0.05).

*Detection according to the nature of the obstacles:* The mean frequencies of cube detection were 55.7 ± 15.3% (EB), 43.0 ± 18.0% (LB), and 50.5 ± 21.8% (SC). The corresponding mean frequencies for postdetection were 32.1 ± 13.2% for EB, 25.9% ± 14.5% for LB and 33.3 ± 7.5% for SC. EB detected 35.1 ± 24.4% of steps, while LB and detected 16 ± 18.3% and 28.9 ± 19.8%, respectively. Doors were the most easily detected obstacles with mean performances of 97.2 ± 13.2% for EB, 95.0 ± 7.5% for LB, and 91.6 ± 10.6% for SC. The ANCOVA corrected for age and sex indicated a significant effect of type of obstacle F(3, 94) = 94.141, *p* < 0.01, η^2^ =0.730), but no group, age, or sex effects, nor for the interaction between groups and types of obstacles (*p* > 0.05). Post hoc tests with Bonferroni correction revealed that doors and cubes were significantly better detected than steps and posts (*p* < 0.001), but cubes were less well detected than doors (*p* < 0.001). No significant differences were found between detection of steps and posts (*p* > 0.05) (Figure 4A).

*Collisions according to the nature of the obstacles:* The collisions rates were about 20% per type of obstacles in each group, except for steps, which had up to 80% collision rates. Indeed, the EB group collided with 68.3 ± 22.8% of steps, but only 16.9 ± 21.5% of cubes, 14.6 ± 14.1% of posts, and 6.3 ± 4.0% of door obstacles. For LB, results were similar with 80.2 ± 17.2% collisions with steps, but only 23.4 ± 16.9% with cubes, 18.6 ± 8.6% with posts, and 8.7 ± 7.9% with doors. In SC, there were 77.5 ± 22.7% collisions with steps, but only 22.8 ± 17.3% for cubes, 17.9 ± 10.9% for posts, and 34.0 ± 22.6% for doors. Much like the detection performances, the ANCOVA corrected for age and sex revealed a significant effect in the type of obstacles (F(2, 94) = 79.811, *p* < 0.001, η^2^ = 0.680), but not in groups, age, or for the interaction between groups and type of obstacles (*p* > 0.05). Post hoc tests with Bonferroni correction indicated that collision with steps was significantly greater than for the other obstacles (*p* < 0.001). There was no significant difference in collision frequencies for cubes, posts, and doors. Moreover, the ANCOVA indicated a significant effect of sex in collision rates (F(1, 94) = 6.132, *p* = 0.015). However, the effect size (η^2^ = 0.017) was very low, and the gender difference did not survive the post hoc Mann–Whitney test (*H*(106) = 1140, *p* > 0.05) (Figure 4B).

## 4. Discussion

The goal of the present study was to investigate the reliability of the suggested EyeCane’s downward sensor by testing a single-sensor EyeCane pointed toward the ground in the presence of obstacles that could induce tripping and falling in daily-life situations. Participants’ performance in our obstacle course should serve as a reasonable indicator of the potential of the device as a primary mobility aid in various circumstances (i.e., indoor and outdoor travel). Furthermore, we wanted to investigate the differences in capacities and strategies between late- and early-blind individuals, to determine their implications for improving the design of the EyeCane and other mobility aids’ design. We hypothesized that participants would have particular difficulty in detecting and avoiding the “step” obstacle, which would thus present a major limitation for the reliance on the EyeCane as a primary mobility aid. We also hypothesized that EB individuals, given their superior abilities with SSDs [13], would show better navigation capacities with the device than LB and SC participants.

The experiment was centered around three navigational tasks—obstacle detection, identification, and avoidance—which require distinct but interlinked processes to sustain safe navigation. In fact, these three processes all require an ability to attribute the device signal to specific objects in the external environment. This phenomenon, called *distal attribution*, is mandatory for navigation using devices such as ETAs and SSDs, and is mainly achieved during the training phase when participants calibrate their internal representation by touching obstacles [33]. In using the EyeCane to execute successfully our navigation task, the user must analyze the vibration intensity to extract information about the obstacle’s presence and position and then analyze the vibration pattern generated by the scanning movement to extract the obstacle’s identity (i.e., form and dimensions) and plan an avoidance path.

### 4.1. Influence of Visual Experience

Overall detection, avoidance, and identification performances were not statistically different between groups. This result might be surprising given the established literature on sensory substitution showing superior abilities of EB [34], but some studies on minimalist SSDs (Guidance-SSD; Sound of Vision) have also observed equivalent performances of EB, LB, and SC [24,35,36]. One might thus suppose that our findings of efficient performances for every group despite only brief training may be an indicator of the device’s simplicity and ease of use [37]. However, both blind groups were significantly better at avoiding the obstacles they detected, while EB were significantly faster than LB and SC at doing the task. A possible explanation for this finding is that the EyeCane provides information on the relative distance between the user and the obstacle, which necessarily places the user at the center of his/her perceived space. This favors the use of an egocentric (body centered) spatial representation, which is known to predominate in EB, whereas LB and SC individuals are more used to working with allocentric (object centered) strategies [38]. Indeed, a normative visuocentric development favors spatial navigation behavior toward the use of an allocentric frame of reference [39]. We cannot exclude another possible explanation for the faster task performance of EB, namely the use of passive echolocation. Indeed, in O&M training, blind individuals are taught to use environmental sounds to obtain spatial information and detect objects [30]. Although we attempted to control for active echolocation by administering the test in a relatively nonechoing sonic and silent environment, we did not ask participants to wear earplugs. In addition, our blindfolding of sighted participants certainly placed them at a disadvantage, eliminating their usual visual inputs impairs their spatial abilities and postural control, even while using an SSD [6]. Therefore, these factors could have influenced the observed group differences in transit times for the obstacle course.

How the lack of visual experience developmentally affects the organization of the visual system and spatial representation could also be a factor in the present findings. Indeed, when deprived of vision early in life, the brain undergoes massive structural and functional reorganization that allows the visual cortex and its associative areas to recruit the other senses, often resulting in superior skills in these modalities (mainly touch and audition) [34]. Superior tactile acuity due to the rerouting of tactile input to the visual cortex in EB [40,41,42] could have facilitated their perception of vibration changes from the device, thus resulting in greater ease in obstacle detection and avoidance, leading to faster crossing times compared to the LB group. Due to these plastic phenomena, we expected EB to be more efficient in detecting the “step” obstacle and its faint vibration changes. However, their performance of this task did not differ significantly from that of the other groups, perhaps due to limitations in subject recruitment and the number of trials. Furthermore, several studies have shown that blind individuals can use the same structures of the navigational network that are normally involved in visually guided navigation (i.e., hippocampal formation, parahippocampal gyrus, posterior parietal cortex, and occipital areas) [10,43,44,45]. This circuit of brain regions seems to be vision dependent in sighted individuals, since it is not recruited when they navigate while blindfolded [10]. Moreover, a recent study with a tactile SSD reported that the EB recruit a sensorimotor circuit (e.g., inferior parietal cortex and areas 3a and 4p) when learning obstacle detection, while the sighted use mainly the medial temporal lobe (e.g., hippocampus and entorhinal cortex) [44]. Such crossmodal reorganization can also be present in the LB brain, albeit to a lesser scale and enhanced tactile acuity has also been observed in such individuals [46]. However, these processes are training dependent and, given that the cortex of LB is initially shaped by vision, the functional reallocation may not occur to the same extent as seen in the EB [47,48]. Although we found no significant effect of BDI on any of the results, this could reflect the considerable heterogeneity of the groups with respect to blindness onset, duration, and cause.

### 4.2. The EyeCane as a Primary Mobility Aid

The results obtained in this study are in line with previous studies on the EyeCane [23,24,25]: our participants were able to detect and avoid a variety of obstacles with good reliability (overall detection and avoidance performance) with little training (15 min). However, this study differed in the way subjects were instructed to scan with the device. Our participants scanned toward the ground with a technique resembling their daily use of the white cane, with the goal of detecting small obstacles or denivelations such as curbs on which they might trip and fall. While this technique allowed the efficient detection of obstacles such as walls and door frames, thin posts, and knee-high obstacles and was successful in sustaining good identification and avoidance performances for such obstacles, participants were less successful in detecting and avoiding the 0.15 m high “step” obstacle. In line with our hypothesis, this can be explained by the very faint vibration amplitude difference between the ground and the top of the “step” obstacle, which is difficult to perceive. A previous study with the EyeCane also using differently sized obstacles likewise demonstrated that bigger obstacles occupying the entire width of the hallway (similar to our “door frame” obstacle) were easier to detect than smaller and lower obstacles (like our “cube”, “step”, and “post”) [26].

Present evidence shows that the EyeCane is an effective tool for detecting high obstacles by providing simple and relevant feedback to the user during mobility, even when the device is pointed toward the ground. However, this instrumentation seems insufficient for properly detecting those obstacles located at ground level (“step” obstacle), which is a mandatory function for safe navigation in both indoor (i.e., stairs) and outdoor (i.e., curbs and potholes) environments. Therefore, our results suggest that the EyeCane in its present state does not suffice as a primary mobility aid but rather can serve as a secondary aid used as an attachment to the white cane. The tactile signal given by the device in addition to the information obtained with the white cane would greatly augment the spatial information the users can acquire, increase their understanding of their surroundings, and thus increase their safety and independence. Indeed, our results show that the EyeCane would likely complement the white cane as an added protection against obstacles higher than the waist but also as a tool to identify (i.e., size, shape, etc.) these looming obstacles, which is crucial information to devise an avoidance strategy.

However, we note that this study took place within an experimental period of only three hours. We can thus hypothesize that with greater training and usage time, participants would likely get better at detecting and avoiding ground level obstacles. Nonetheless, given the absence of physical contact with the ground (normally provided by the tip of the white cane), it is unlikely that a long-term user of the EyeCane alone would reach the proficiency of a white cane user. Indeed, the sensors used in the device have certain limitations with regard to their sensitivity to environmental conditions and accuracy.

### 4.3. Implications for ETA Design

This study tested the reliability of a single-sensor EyeCane pointed toward the ground. Since our experimental design allowed to test its reliability to detect ground obstacles as well as higher obstacles, our results show that this device as a sole aid would not be sufficient to support safe navigation. Here, we list three significant limitations not only for the single-sensor EyeCane but also for the suggested three-sensors EyeCane (Figure 1D) [26], with general implications for the design of ETAs:(1)*Loss of physical contact with the ground*. The white cane contact assures a high level of reliability in detecting ground obstacles and drop-offs such as curbs [49], and its replacement with the present downward sensor leads to less confidence in detecting drop-offs and ground obstacles.(2)*Added complexity*. While the EyeCane’s main advantage has always been its simplicity, ease of operation, and intuitive feedback, adding multiple sensors would add complexity and significantly increase the required training time. Indeed, a three-sensor design would necessitate feedback of a different nature (i.e., different modalities or frequency coding) for each sensor, as they each provide different spatial information that must be discriminated by the user.(3)*User identification*. As the white cane allows pedestrians and drivers to identify the blind user and to assure his/her safety [3], a three-sensor EyeCane might lead to the loss of user identification and impede safety.

Despite these design limitations, the EyeCane (with a single infrared sensor) has proved its reliability in obstacle detection and avoidance in multiple studies [23,26,50]. With its advantages with respect to ease-of-use, simple feedback, light weight, and long battery life [23], the EyeCane may prove to be most useful as an attachment to white cane to improve the individual’s safety, and reduce the risk of head injury [6]. In fact, all primary ETAs incorporate sensors attached to a physical cane (i.e., ultracane and WeWalk cane) [51,52], since blind individuals habitually rely heavily on the white cane’s physical contact with the environment, as shown in previous studies on mobility aids [20,36]. However, these devices are often clumsy and expensive [53]. Thus, the low cost and ergonomic light weight, design of the EyeCane presents it as an affordable alternative for those who cannot afford an “all-in-one” electronic cane.

Further work should investigate the reliability and user acceptance [54] of a slim, ergonomic EyeCane fitted to the grip of a modified white cane’s grip and equipped with an upward sensor. According to our findings, such an attachment would result in better obstacle identification and would substantially increase safety and route planning in the face of an environment encumbered by obstacles. We suppose that such a hybrid design would have great potential for adoption in the blind and visually impaired community worldwide.

## 5. Conclusions

This study aimed to investigate the EyeCane’s potential as a primary mobility aid to avoid obstacles and diminish the risk of injuries and falling. Our results showed that the EyeCane efficiently enabled the users to detect, identify, and avoid large and high obstacles but failed to provide efficient coverage for obstacles at ground level. Indeed, removing the white cane and its physical contact with the ground leads to inconsistent detection of ground obstacles (i.e., steps, curbs, and holes) and higher risks of tripping and falling. Thus, the EyeCane is a potentially beneficial and low-cost attachment to the white cane that can significantly improve the individual’s safety during mobility by providing coverage to obstacles above the waist.

## Figures and Tables

**Figure 2 sensors-21-02700-f002:**
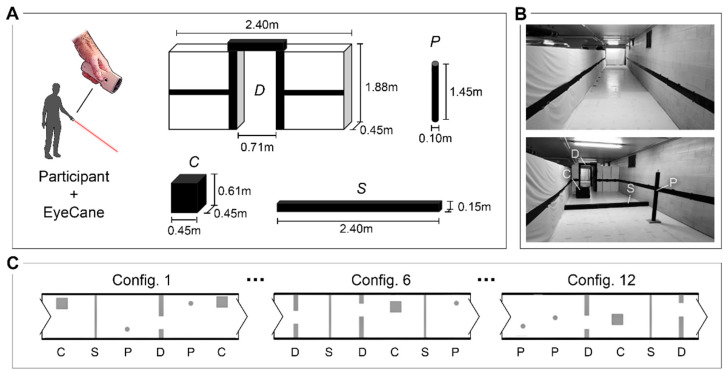
The experimental setup and procedure. (**A**) A participant handling the EyeCane in a downward directed manner; four type of obstacles: D= door, P = post, C = cube, and S = step. (**B**) At the top, a photograph of the empty obstacle course and at the bottom an example of obstacle placement. (**C**) Three schematic representations of the twelve different trial configurations.

**Figure 3 sensors-21-02700-f003:**
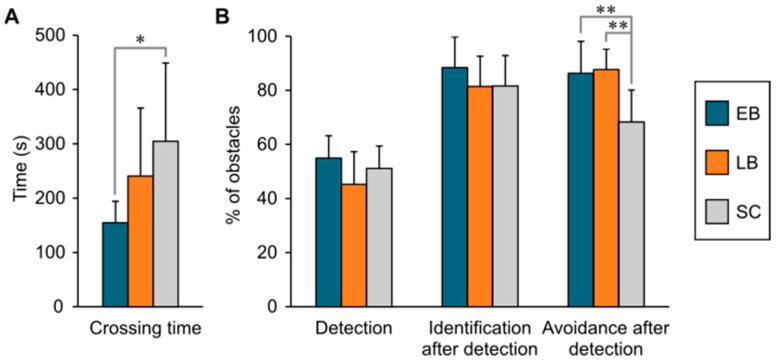
The average crossing time (**A**), and the mean rates of detection, identification, and avoidance by the three subject groups (**B**). Significant differences are indicated by asterisks (* = *p* < 0.05; ** = *p* < 0.01). EB = early blind; LB = late blind; and SC = sighted control.

**Figure 4 sensors-21-02700-f004:**
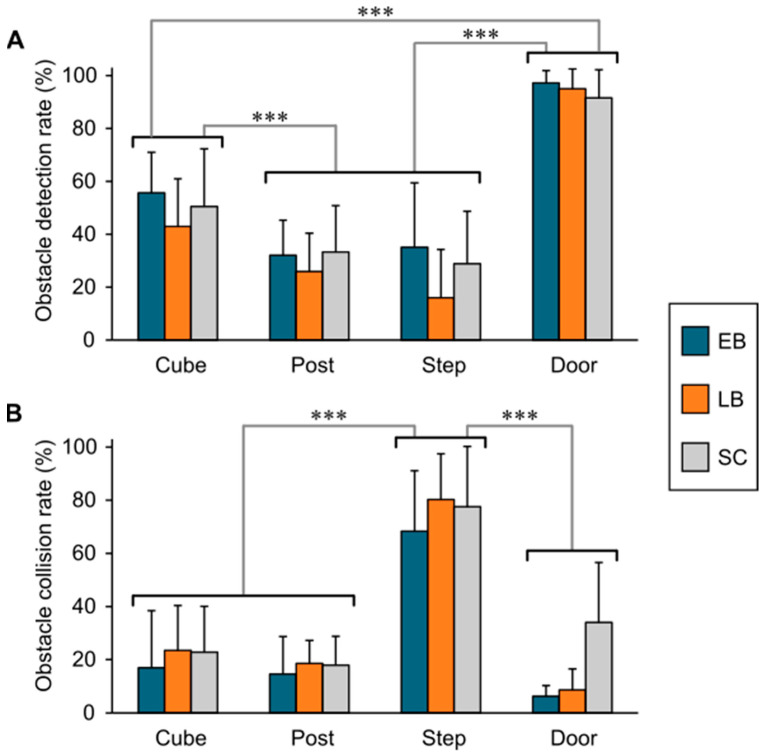
The average rates of obstacle detection (**A**) and collision (**B**). Significant differences are indicated by asterisks (*** = *p* < 0.001). EB = early blind; LB = late blind; and SC = sighted control.

**Table 1 sensors-21-02700-t001:** Blind participants’ characteristics.

Participant	Age	Sex	Age of Onset of Blindness	Blindness Duration Index	Cause of Blindness	Residual Perception
LB1	55	F ^1^	24	0.56	Retinitis pigmentosa	yes
LB2	25	M ^2^	17	0.32	Retinitis pigmentosa	-
LB3	70	M	38	0.46	Meningitis	-
LB4	38	F	20	0.64	Retinal cancer	-
LB5	46	M	40	0.13	Meningitis	-
LB6	56	F	20	0.47	Retinal cancer	-
LB7	47	F	22	0.53	Diabetic retinopathy	-
LB8	44	F	17	0.61	Glaucoma	-
LB9	59	F	57	0.03	Retinitis pigmentosa	yes
EB1	48	M	Perinatal	-	Retinopathy of prematurity	-
EB2	33	M	Perinatal	-	Retinopathy of prematurity	-
EB3	63	M	Perinatal	-	Retinopathy of prematurity	-
EB4	54	M	Perinatal	-	Retinopathy of prematurity	-
EB5	56	M	Perinatal	-	Retinopathy of prematurity	-
EB6	36	M	Perinatal	-	Retinopathy of prematurity	-
EB7	31	M	Perinatal	-	Retinopathy of prematurity	-
EB8	40	M	Perinatal	-	Retinopathy of prematurity	-
EB9	33	F	Perinatal	-	Retinopathy of prematurity	-
EB10	51	M	Perinatal	-	Meningitis	-

^1^ Female, ^2^ Male.

## Data Availability

Data are available at ismael.djerourou@umontreal.ca.

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
