# Peer review of "Blindness and the Reliability of Downwards Sensors to Avoid Obstacles: A Study with the EyeCane"

_sensors, 2021, doi:10.3390/s21082700_

Round 1

Reviewer 1 Report

This is an interesting contribution aimed at studying the reliability of the EyeCane aid.

Also, the topic is very interesting and attractive for the community.

The overall organization is good and the English writing clear.

Nevertheless, I have some comments that should be addressed before publication.

From the introduction, it seems that only ETAs based on infrared exist, and marginally we have camera and ultrasonic systems. However, there also additional emerging technologies which have been extensively studied in recent years. The main example is represented from the electromagnetic/radar technology. Some of the references reported hereafter concern this topic but additional papers focused on microwave/millimeter wave radars employed as ETAs can be found in the literature.

Moreover, there also valuable reviews/papers from M. M. Islam’s group, giving an interesting overview of the different existing technologies.

Some insights present in the following references might be very beneficial to increase the paper scientific soundness and to provide a different point of view to the readers.

- V. Di Mattia et al., "A feasibility study of a compact radar system for autonomous walking of blind people," 2016 IEEE 2nd International Forum on Research and Technologies for Society and Industry Leveraging a better tomorrow (RTSI), Bologna, Italy, 2016, pp. 1-5, doi: 10.1109/RTSI.2016.7740599.

-Cardillo, E.; Caddemi, A. Insight on Electronic Travel Aids for Visually Impaired People: A Review on the Electromagnetic Technology. Electronics 2019, 8, 1281. https://doi.org/10.3390/electronics8111281.

-E. Cardillo et al., "An Electromagnetic Sensor Prototype to Assist Visually Impaired and Blind People in Autonomous Walking," in IEEE Sensors Journal, vol. 18, no. 6, pp. 2568-2576, 15 March15, 2018, doi: 10.1109/JSEN.2018.2795046.

-M. M. Islam, M. Sheikh Sadi, K. Z. Zamli and M. M. Ahmed, "Developing Walking Assistants for Visually Impaired People: A Review," in IEEE Sensors Journal, vol. 19, no. 8, pp. 2814-2828, 15 April15, 2019, doi: 10.1109/JSEN.2018.2890423.

-M. M. Islam, M. S. Sadi and T. Bräunl, "Automated Walking Guide to Enhance the Mobility of Visually Impaired People," in IEEE Transactions on Medical Robotics and Bionics, vol. 2, no. 3, pp. 485-496, Aug. 2020, doi: 10.1109/TMRB.2020.3011501.

It might be useful and interesting to add some details concerning the energy consumption (current/power consumption, weight of the battery etc).

Is your system reliable to different light conditions and to surfaces with different roughness?

Author Response

Reviewer #1

Comment 1. This is an interesting contribution aimed at studying the reliability of the EyeCane aid.

Also, the topic is very interesting and attractive for the community.

The overall organization is good and the English writing clear.

Response: We thank the reviewer for his/her positive perspective on our work.

Comment 2. Nevertheless, I have some comments that should be addressed before publication.

From the introduction, it seems that only ETAs based on infrared exist, and marginally we have camera and ultrasonic systems. However, there also additional emerging technologies which have been extensively studied in recent years. The main example is represented from the electromagnetic/radar technology. Some of the references reported hereafter concern this topic but additional papers focused on microwave/millimeter wave radars employed as ETAs can be found in the literature.

Moreover, there also valuable reviews/papers from M. M. Islam’s group, giving an interesting overview of the different existing technologies.

Some insights present in the following references might be very beneficial to increase the paper scientific soundness and to provide a different point of view to the readers.

- V. Di Mattia et al., "A feasibility study of a compact radar system for autonomous walking of blind people," 2016 IEEE 2nd International Forum on Research and Technologies for Society and Industry Leveraging a better tomorrow (RTSI), Bologna, Italy, 2016, pp. 1-5, doi: 10.1109/RTSI.2016.7740599.

-Cardillo, E.; Caddemi, A. Insight on Electronic Travel Aids for Visually Impaired People: A Review on the Electromagnetic Technology. Electronics 2019, 8, 1281. https://doi.org/10.3390/electronics8111281.

-E. Cardillo et al., "An Electromagnetic Sensor Prototype to Assist Visually Impaired and Blind People in Autonomous Walking," in IEEE Sensors Journal, vol. 18, no. 6, pp. 2568-2576, 15 March15, 2018, doi: 10.1109/JSEN.2018.2795046.

-M. M. Islam, M. Sheikh Sadi, K. Z. Zamli and M. M. Ahmed, "Developing Walking Assistants for Visually Impaired People: A Review," in IEEE Sensors Journal, vol. 19, no. 8, pp. 2814-2828, 15 April15, 2019, doi: 10.1109/JSEN.2018.2890423.

-M. M. Islam, M. S. Sadi and T. Bräunl, "Automated Walking Guide to Enhance the Mobility of Visually Impaired People," in IEEE Transactions on Medical Robotics and Bionics, vol. 2, no. 3, pp. 485-496, Aug. 2020, doi: 10.1109/TMRB.2020.3011501.

Response: We thank the reviewer for this comment, we therefore added these references and mentioned more ETA technologies.

Location of change: Lines 70-72, 78, 79

Comment 3. It might be useful and interesting to add some details concerning the energy consumption (current/power consumption, weight of the battery etc).

Response: We followed the reviewer’s suggestion and added specifications on the device such as dimensions, weight, and battery life.

Location of change: lines 142-144.

Comment 4. Is your system reliable to different light conditions and to surfaces with different roughness?

Response: We thank the reviewer for this question, we therefore specified that the device has been tested on different material and lighting conditions.

Location of change: lines 151, 152

Reviewer 2 Report

The manuscript presents an experimental study on electronic travel aid EyeCane. The paper is based on statistical data from a group of EyeCane users with different obstacle configurations. The study is well designed and has a clear research goal.

The title is informative and relevant. The abstract match the rest of the article.

The background study is well prepared, the authors are well informed about the latest trends in the field, and have a broad knowledge of the studied topic. The research question is clearly outlined, and the study methods are well presented.

In my opinion, it would be good to have current EyeCane configuration parameters with a specification of the sensors used and the type of feedback for different obstacles. That information may be useful for the researchers who would like to repeat the study or continue developing in that field.

The data presented appropriately. Tables and figures are seeming to be relevant and apparently presented. It is clear what are the practically meaningful results of the work.

As a weakness of the study, I would highlight missing the conclusion section. Please try to summarize the most critical findings in the study.

Overall, I suggest accepting the manuscript to be a part of the MDPI Sensors journal after minor revision and adding the tool's description (EyeCane) and proper conclusion.

Author Response

Reviewer #2:

Comment 1. The manuscript presents an experimental study on electronic travel aid EyeCane. The paper is based on statistical data from a group of EyeCane users with different obstacle configurations. The study is well designed and has a clear research goal.

The title is informative and relevant. The abstract match the rest of the article.

The background study is well prepared, the authors are well informed about the latest trends in the field, and have a broad knowledge of the studied topic. The research question is clearly outlined, and the study methods are well presented.

Response: We thank the reviewer for his/her positive perspective on our work.

Comment 2. In my opinion, it would be good to have current EyeCane configuration parameters with a specification of the sensors used and the type of feedback for different obstacles. That information may be useful for the researchers who would like to repeat the study or continue developing in that field.

Response: We followed the reviewer’s suggestion and added information about how participants can identify obstacles by scanning them with the device. No configurations are needed.

Location of change: lines 187-189.

Comment 3. The data presented appropriately. Tables and figures are seeming to be relevant and apparently presented. It is clear what are the practically meaningful results of the work.

Response: We thank the reviewer for his/her positive perspective on our work.

Comment 4. As a weakness of the study, I would highlight missing the conclusion section. Please try to summarize the most critical findings in the study.

Response: We thank the reviewer for this comment, we therefore added a Conclusion paragraph at the end of the manuscript

Location of change: Lines 471-480

Reviewer 3 Report

The manuscript "Blindness and the reliability of downwards sensors to avoid obstacles: a study with the EyeCane" (sensors-1179529), investigate the potential of the EyeCane (electronic travel aid with two IR distance sensors to perform one line measurement) as a primary aid for spatial navigation. However, in its current form, the manuscript has several flaws that must be eliminated:

The developed system (EyeCane) consists of two IR distance sensors for measuring the distance in one direction. A sensor (Sharp GP2D12) measures from 6 cm to 80 cm. The other sensor (Sharp GP2Y0A710) measures from 100 cm to 550 cm. There will be a blind area, between 80 cm and 100 cm (20 cm). How was it solved? Won't this discontinuity influence the detection of some obstacles?

I noticed that, Fig. 2 B, all obstacles are completely painted black and / or have a black area. Because? Given that they are zones of non-reflection of IR light, is this not helping to detect obstacles by creating transition zones? These conditions are hardly to be found in a real environment. What is their influence?

In addition to these major edits, the author should consider these minor suggestions:

  • In line 254 is missing a point in the number 005 (suppose it is 0.05) .
  • In line 343 ‘the’ shouldn't be there – “One might thus suppose that our findings …”

Author Response

Reviewer #3:

Comment 1. The developed system (EyeCane) consists of two IR distance sensors for measuring the distance in one direction. A sensor (Sharp GP2D12) measures from 6 cm to 80 cm. The other sensor (Sharp GP2Y0A710) measures from 100 cm to 550 cm. There will be a blind area, between 80 cm and 100 cm (20 cm). How was it solved? Won't this discontinuity influence the detection of some obstacles?

Response: We thank the reviewer for this comment, we corrected the specifications of the device used in the study. The Eyecane’s iteration used in our study is equipped with a Sharp GP2Y0A02YK0F sensor which has a 20 to 150 mm detection range. Therefore, there is no blind spot in the device’s coverage within these values.

Location of change: Lines 145, 148

Comment 2. I noticed that, Fig. 2 B, all obstacles are completely painted black and / or have a black area. Because? Given that they are zones of non-reflection of IR light, is this not helping to detect obstacles by creating transition zones? These conditions are hardly to be found in a real environment. What is their influence?

Response: The obstacles were painted black or partly black as the obstacle course used in this study was also used for experimentations with the tongue display unit (TDU), for reference see Chebat et al., 2011. The TDU delivers a black and white image in the form of electrical stimulations on the tongue. Therefore, during experiments, obstacle courses are designed with black obstacles on white background, because it offers the optimal contrast. The black stripes on the walls (and doors) were designed similarly as the accessibility measures in Montreal (URL: https://ville.montreal.qc.ca/pls/portal/docs/page/d_social_fr/media/documents/au_batiments_muni_2017.pdf), since walls must be distinguished to avoid collisions. Such design choice did not have any incidence on the ability of participants to detect obstacles with the EyeCane as all participants were able to detect, scan and judge obstacles’ size and distance during the training phase and experiment.

Chebat D-R, Schneider FC, Kupers R, Ptito M. Navigation with a sensory substitution device in congenitally blind individuals. Neuroreport. 2011;22:342-7.

Comment 3. In addition to these major edits, the author should consider these minor suggestions:

  • In line 254 is missing a point in the number 005 (suppose it is 0.05) .
  • In line 343 ‘the’ shouldn't be there – “One might thus suppose that our findings …”

Response: We thank the reviewer for this comment, we therefore corrected these details

Location of change: Lines 258, 347

Round 2

Reviewer 1 Report

The author addressed all my concerns.

Reviewer 3 Report

I have no further comments. All the answers given were satisfactory.